# Attributable Mortality to Candidiasis in Non-Neutropenic Critically Ill Patients in the ICU and a Post-Mortem Study

**DOI:** 10.3390/jof11120871

**Published:** 2025-12-08

**Authors:** Jordi Ibañez-Nolla, Felip Garcia, Miquel A. Carrasco, Miquel Nolla-Salas

**Affiliations:** 1Health Sciences Faculty, Blanquerna-Universitat Ramon Llull, 08025 Barcelona, Spain; 2Medicine Department, Fundació Hospitalaries Barcelona, 08035 Barcelona, Spain; 3Pathology Department, Hospital Quirón Barcelona, 08023 Barcelona, Spain; 16965fgh@comb.cat; 4Pathology Department, Hospital General de Catalunya-Quirón, 08195 Sant Cugat del Vallés, Spain; mcarrasco@quironsalud.es; 5Unió Catalana d’Hospitals, 08009 Barcelona, Spain; mnollasalas@gmail.com

**Keywords:** candidiasis, attributable mortality, non-neutropenic, ICU, post-mortem study

## Abstract

Candidiasis remains one of the most challenging infections to treat in critical care, due to its diagnostic difficulties and uncertainty regarding whether it can be directly related to the death of patients with multiorgan failure. This study aims to verify that the statistically attributable mortality in this infection is as consistent as the post-mortem attributable mortality. A prospective study was conducted in non-neutropenic ICU patients in whom *Candida* was detected. Invasive candidiasis is defined based on evidence of disseminated or multifocal candidiasis. Post-mortem study is used as the gold standard for *Candida*-attributable mortality, and is compared with attributable mortality determined according to clinical study and statistically attributable mortality in relation to the overall mortality of ICU patients and colonized patients. The post-mortem attributable mortality was 30.6% and 22.6% according to the clinical study, while the statistically attributable mortality was 25% in relation to overall ICU mortality and 27% in relation to *Candida* colonization. Thus, the results of the different calculations of attributable mortality (statistical vs. crude death rate) due to *Candida* are in agreement. The use of this metric may help to improve ICU outcomes for non-neutropenic critically ill patients with candidiasis.

## 1. Introduction

*Candida* is part of the endogenous human flora that can become invasive. It can proliferate uncontrollably when the patient has an immunological deficit, causing inflammation and ulceration in normal growth sites such as the mouth, vagina, skin, or gastrointestinal tract. From there, it can spread through the hematogenous route, affecting internal organs and giving rise to serious disease. Three factors have been identified in relation to risk of dissemination: change in endogenous flora due to long-term use of antibiotics, which causes excessive growth of *Candida*; significant compromise of the host’s defenses; and introduction of the organism intravenously. In the clinical context, unfortunately, signs of dissemination may be absent or may present too late to be useful. Positive blood or urine cultures are present in less than half of patients, and candidemia frequently appears for the first time only a few days before death [1].

Forty years after the initial transmission of this knowledge, the mortality rate due to disseminated candidiasis in the ICU remains higher than 50%, as indicated in the Consensus recommendations published in 2024 [2].

Although a wide variety of antifungals are currently available to treat this infection, these patients’ prognosis has not been improved.

Different concepts are used regarding candidiasis in the ICU: *Candida* colonization (CC), characterized by isolation of yeast from a single biological sample obtained from surfaces (skin, mucous membranes, or fluids considered *non-sterile*); multifocal candidiasis (MC), characterized by isolation of *Candida* in more than one sample of mucous membranes or *non-sterile* fluids obtained simultaneously and sterilely [3]; invasive candidiasis (IC), characterized by isolation in more than one blood culture or demonstration of deep-seated candidiasis in a single site without evidence of contamination of the site [2]; and disseminated candidiasis (DC), characterized by deep-seated candidiasis demonstrated in more than one organ or sterile fluid outside the blood.

Although consensus on candidiasis in the ICU includes MC in the CC group, since we determined that patients with multiorgan dysfunction and long-term ICU stay with MC are at high risk of DC, MC was equated with IC. The same antifungal therapeutic protocol was applied to both MC and IC. Consequently, MC is excluded from the CC group, whose members should not receive prophylactic or therapeutic antifungals [3]. When an ICU patient with multiorgan dysfunction dies during the course of candidiasis, it is necessary to conduct an evaluation of whether the death was directly attributable to *Candida* and, if so, whether it was due to diagnostic delay that led to inadequate treatment. This study focuses on the first point, with the aim of developing a reliable metric to define death attributable to *Candida*.

To add value to the data presented in studies on *Candida* infections in the ICU, the concept of attributable mortality to *Candida* is considered more useful than the crude death rate. Attributable mortality is an important metric that mirrors the public health effect of a potentially harmful infection by accounting not only for the effect of the infection on mortality, but also its prevalence within the target population [4]. The crude death rate is a basic demographic indicator that does not consider the particularities of the population analyzed.

The purpose of this study is to present the utility of attributable mortality for this type of infection in the ICU and to compare the gold standard of post-mortem attributable mortality with clinical and statistically attributable mortality.

## 2. Materials and Methods

Patients: A prospective study was conducted over 7 years, before the COVID-19 pandemic, in a 10-bed general ICU. Inclusion criteria were ICU patients with ICU stay > 1 week and multiorgan disfunction (MODS), a SOFA Score > 4, and signs of sepsis. The diagnostic algorithm (Figure 1) was applied after the isolation of *Candida* spp. in any sample for the study of sepsis or isolation in the post-mortem study. Exclusion criteria were isolation of *Candida* spp. in cultures prior to ICU admission; severe neutropenia (<500/mm^3^); and patients in whom it was not possible to conduct a study of the presence of yeast in different *foci* during their ICU stay once it was detected in a sample. Following the detection of DC through samples considered *sterile* (cerebrospinal, pleural, or pericardial fluids; peritoneal or bile fluids), histological samples from deep organs, endophthalmitis, or candidemia with negative line tip cultures, antifungal treatment was immediately started.

Screening consisted of obtaining samples of bronchial secretions, urine, throat smear, stool, wound, drainage fluids, fluids from sterile areas (cerebrospinal fluid, pleura, peritoneum, biliary, and pericardium), histological samples, and blood cultures through venipuncture and cultures from catheter tips. The study algorithm also included: 1. Cutaneous tests to evaluate cellular immunity [5,6], utilizing Multitest IMC® (Rhône Poulenc Pharma SAE, Alcorcón, Spain). 2. Ophthalmological evaluation to rule out endophthalmitis [7].

Clinical, microbiological, and demographic data were collected from the study group. A post-mortem study involving microbiological and histological analyses was carried out when family consent was given, in line with the hospital’s Ethics and Mortality Committees guidelines. For all patients who died in the ICU, regardless of whether *Candida* spp. were isolated, their relatives were asked for consent to conduct a post-mortem study.

According to the results of the diagnostic algorithm, the patients were classified into the following groups: CC when isolation occurred in a single focus not suggestive of dissemination or at the skin level, feces, or vaginal smear; IC when they meet criteria for multifocal, invasive, or disseminated Candidiasis [8].

In all cases included in the study—both those considered as IC and CC—*Candida* follow-up cultures were performed every 5–7 days after screening until discharge from the ICU.

Microbiological study: Regardless of whether the samples were obtained pre- or post-mortem, cultures were made in media for both bacteria (blood-agar) and fungi (Sabouraud with gentamicin and chloramphenicol). They were incubated at 37 °C under aerobic and anaerobic conditions and in an atmosphere enriched with CO_2_ [9].

All samples were obtained under maximum aseptic conditions either by direct puncture, pharyngeal smear, gastric aspirate, or bronchoalveolar aspirate or lavage. In the case of blood cultures, three samples were obtained through direct venipuncture. Maki’s semiquantitative culture technique was applied to study blood catheters.

Post-mortem samples were collected within the first 24 h of death, both for cultures and histological study. Pending the collection of post-mortem samples, the corpses were kept at 4 °C. The specimens were obtained using non-sterile instruments and sent to the laboratory, where the capsular surfaces were seared by heat and then tissue samples were obtained for culture. The De Jongh method modified by Dolan [10] was used. The congruence or non-congruence of pre- and post-mortem positive cultures was analyzed to validate post-mortem cultures. Identification of the same microorganisms in post-mortem cultures as in pre-mortem cultures or negative post-mortem cultures in a correctly treated patient was considered congruent; identification of microorganisms in post-mortem cultures not identified pre-mortem was considered non-congruent.

When the study began, our center did not yet have DNA analysis methods to identify the species and biotypes of *Candida* spp. Instead, methods based on biochemical tests, fermentation, growth inhibition, and enzymatic and immunological identification with specific antisera were used.

Post-mortem study: We aimed to conduct post-mortem studies on at least 50% of the deceased individuals in the study population. All autopsies were performed according to the M. Lefulle technique [11], assessing the causes of death and secondary injuries. A histological study was carried out during the autopsy to search for microorganisms. Usual staining techniques (hematoxylin–eosin, periodic acid–Schiff (PAS), and Gomori’s silver methenamine) were systematically performed. These techniques allowed us to identify yeast in filamentous forms (*pseudohyphae*) and/or spores associated with inflammatory infiltrates [12,13]. Samples for post-mortem cultures were collected from different organs (e.g., lung, spleen) and organic fluids (e.g., pleural, peritoneal, cerebrospinal fluid, abscesses, purulent urine), according to the clinical data and the macroscopic findings of the autopsy [10,14,15,16]. We assessed yeasts according to the affected organs; the findings of the post-mortem study, using both histology and cultures, were interpreted according to the samples’ organs of origin [13,16,17,18].

Positive Samples and Syndrome Directly Related to Death: The culture was considered significant when the clinical symptoms (e.g., respiratory failure, sepsis or Multiple Organ Dysfunction Syndrome (MODS), non-septic shock, liver failure, or brain death) were directly related to the patient’s death and concurred with the results of positive histological or culture samples:(a)Heart, liver, spleen, and brain: The presence of yeast in histology is associated with polymorphonuclear inflammatory *foci* in different areas of the parenchyma studied.(b)Blood and spleen: Cultures were obtained by puncturing cardiac cavities or splenic parenchyma with a sterile needle and prior cauterization of the puncture point.(c)Intestine: The presence of *pseudohyphae or hyphae* in histology that extend to the muscle and/or serosa, together with the presence of an important polymorphonuclear infiltrate. The culture was assessed from samples obtained from the peritoneal cavity or bile fluid.(d)Lung: Histology is evaluated if *pseudohyphae* and spores are found in the parenchyma, alveoli, and vessels, along with polymorphonuclear infiltrates. The culture was obtained from areas of the parenchyma distant from the main bronchi and trachea.(e)Kidneys: The presence of yeast with a polymorphonuclear inflammatory component in the kidney parenchyma. The culture was obtained from areas of the parenchyma and not from the urinary tract.(f)When the culture tests positive for yeasts in isolation form in the trachea, main Bronchi, or urine obtained by bladder puncture, it is considered an indicator of colonization in the post-mortem study, but not of IC.

Positive samples and death attributable to yeast: Post-mortem results were interpreted as follows in relation to yeast as a cause of death:Death was attributable to yeast when
Cultures and histology were positive for yeast;Negative cultures and positive histology for yeast were observed in more than one organ;Negative histology and positive cultures for yeast were observed in significant organs, without identification of other microorganisms.Yeast was not considered as a cause of death when
Cultures and histology were negative for yeast;Negative cultures and positive histology for yeast were observed in a single organ, such as the liver, spleen, intestine, kidney, or lungs;Negative histology and cultures positive for yeast were observed in significant organs, with identification of other microorganisms that may be considered responsible for death.


Antifungal treatment: At the time of the study, only Fluconazole and Amphotericin-B were available as antifungal treatments, which were used in patients classified as having IC. The indicated treatment was fluconazole at a dose of 400 mg/24 h for 10 days. As a second choice, in cases of poor clinical evolution with persistence of *Candida* spp. in the cultures, or when *Candida glabrata* or *Candida krusei* was identified, amphotericin-B was used at a dose of 50 mg/24 h for 10 days. Treatment courses of fewer than five days were considered inappropriate. 

Definition of attributable mortality: Attributable mortality is identified according to the following:Attributable mortality according to the post-mortem study: This is the resulting analysis of deceased patients who, on post-mortem study, show death attributable to yeast and to respiratory failure, sepsis, or Multiple Organ Dysfunction Syndrome (MODS). The “gold standard” for the diagnosis of invasive candidiasis is biopsy [19] or post-mortem study [13].Attributable mortality according to clinical study: This includes deaths in which a post-mortem study was not performed. The group of patients with a moderate probability of death attributable to yeast is defined based on the clinical cause of death, the existence of multifocal yeast isolates, and incomplete antifungal treatment (<5 days) before death.Crude mortality rate: This refers to the attributable mortality resulting from the sum of the cases defined as attributable mortality from the post-mortem study and those defined as attributable mortality from the clinical study.Statistical definition of attributable mortality [20,21,22]: This represents the difference between the mortality of the studied group (IC acquired in the ICU in non-neutropenic patients) and the overall mortality in the ICU.

Statistical analyses: Categorical variables were compared between two groups with the Chi-square test or Fisher’s exact test, as appropriate. Continuous variables were analyzed using Student’s *t*-test or the Mann–Whitney U test when the distribution departed from normality. Results are presented as the mean (standard deviation) or median (range), respectively. To estimate the odds ratio (OR) of attributable mortality for variables of interest with a Confidence Interval (CI), we used a logistic regression model adjusted for potentially confounding variables [23]. Statistical significance was established at *p*-value < 0.05, and data were analyzed using the SPSS statistical program (IBM SPSS version 31).

Study limitations: This study was conducted before the commercialization of new antifungals, such as echinocandins. Thus, the therapeutic results cannot be used in accordance with current therapeutic parameters. A multicenter study was ruled out because the ICUs experienced difficulties in achieving the proposed volume of post-mortem studies. This decision limited the statistical power of the study.

## 3. Results

Recruitment and identification of excluded cases: During the 7-year study period, 3389 patients were treated in the ICU, with an average stay of 5.8 days (1–112) and a mortality rate of 9.6% (324). Of the total number of deaths, a post-mortem study was carried out in 42% of cases (136). During the study period, there were 149 patients (4.4%) in whom yeasts were isolated throughout their stay in the ICU. Following the exclusion criteria set forth, eight cases were rejected. Two were excluded for having isolated *Candida* spp. in cultures prior to admission to the ICU, and both died. The other six were excluded because multifocality study was not performed despite yeasts being isolated in the urine culture, and they all survived without any antifungal treatment. Data from 141 patients were collected, and 4 patients (3%) were subsequently readmitted. *Candida* spp. was re-isolated during treatment in their readmission. A total of 145 cases met the inclusion criteria for the study (Figure 2). The ICU mortality rate was 35% (51/145), and hospital mortality was 46% (67/145). A post-mortem study was performed in 36 of the 51 cases who died before ICU discharge (71%).

The population studied: A total of 77% patients were men (111/145), with an average age of 53 (22) years and an average length of stay in the ICU of 28 days (2–93). A total of 63% of patients (91) were admitted for medical reasons and 37% (54) for surgical reasons. The APACHE III score was 76 (18–138), and the comorbidity rate was 70% (102). The most frequent processes were as follows: chronic obstructive pulmonary disease was observed in 23% (34), diabetes mellitus in 22% (32), arterial hypertension in 22% (32), solid neoplasia in 13% (19), malignant hematological disease in 5% (7), leukopenia with a value of less than 3000/mm^3^ upon admission to the ICU without criteria for neutropenia (>500 neutrophils/mm^3^) in 4% (6), and acquired immunodeficiency syndrome in 3% (4). Risk factors for fungal infection were detected in 94% (139) of the patients upon admission to the ICU and in 100% at the time of the first yeast isolation (Table 1). In 58% (84) of cases, there were a minimum of 10 associated risk factors when the first yeast isolation was obtained.

Fungal infection:

Positive samples: The samples that were most frequently positive—both in the first positive culture and during screening—were the bronchial secretion cultures (66% and 86%, respectively).

Blood cultures: The percentage of patients with candidemia in relation to the number of patients treated during this period was 0.5% (18/3389). In total, there were 24 positive blood cultures in 18 patients. The isolation was obtained in the first sample in 4 patients, during screening in 14, and during the follow-up blood cultures in 6. A total of 27 catheters with colonized tips were detected, including 4 in the first sample, 16 during screening, and 7 in the follow-up cultures. The association of a positive catheter with a positive blood culture was verified in eight patients (44% of patients with a positive blood culture).

Identified yeast species: The most frequently isolated yeast was *Candida albicans*—both in the first positive culture and during screening (80% and 87%, respectively)—followed by *Candida glabrata*, also in both cases (11% and 18%, respectively).

Classification according to location of the yeasts: The patients were categorized into the IC group in 120 cases (83%) [MC in 89 cases (61%) and DC in 31 cases (22%)] and the CC group in 25 cases (17%). IC represented 3.5% of the population treated in the ICU during the study period (120/3389). *Candida* spp. was located according to these groups as follows: MC—The respiratory focus was identified in 96% of patients (85), the digestive focus in 93% (83), the urinary focus in 42% (37), and drainage in 2% (2). Two simultaneous foci were demonstrated in 67% of patients (60), three *foci* were demonstrated in 33% (29), and there were no cases with four *foci.* DC—Patients in this group were classified as such because they demonstrated the presence of endophthalmitis in three cases (including one with a positive blood culture). Positive blood cultures and a negative catheter tip served as evidence in seven cases. The yeast was identified in samples obtained from puncture of abscesses in 12 cases, from cerebrospinal fluid in 2 cases, and from tissue biopsies in another 2 cases. In five cases, evidence of dissemination was demonstrated in a post-mortem study. Yeasts were identified in bronchial secretions (15 cases); the pharynx (15 cases); stool culture (10 cases); gastric aspirate (8 cases); urine culture (5 cases); the vagina (2 cases); wounds (2 cases); catheters (2 cases); catheter together with positive blood culture (1 case); skin (1 case); and the nasal cavity (1 case).

Personalization of antifungal treatment: Antifungal treatment was carried out in 109 cases (75%). In 95 of them, azole derivatives were used for 10 days (1–30), and in 42, amphotericin B was used for 15 days (4–39). In four cases, the administration of amphotericin-B had to be stopped due to nephrotoxicity. In 26 cases, the antifungal was changed due to poor clinical or microbiological response (isolation of *Candida glabrata* or *Candida krusei*). Eleven patients (9.2%) did not receive antifungal treatment or it was inappropriate.

Mortality: The mortality rate was 51% (61/120) in cases of IC [disseminated candidiasis was 48% (15/31) and multifocal candidiasis was 52% (46/89)] and 24% (6/25) in cases of colonization, while the mortality rate for cases with positive blood culture was 39% (7/18). In the group that died in the hospital, 53.7% (36/67) of patients received appropriate antifungal treatment, a post-mortem study was carried out in 50% of cases (18/36), and 46.3% of cases (31/67) did not receive antifungal treatment or it was inappropriate, with a post-mortem study being carried out in 58% of them (18/31).

The mortality rate with appropriate antifungal treatment was 33% (36/109), while mortality in patients without antifungal treatment or with inappropriate treatment was 86.1% (31/36).

Post-mortem study: Although 38 autopsies were performed on these deceased patients, two were excluded for not following the post-mortem study protocol. Among the 36 patients who underwent autopsy, the symptoms directly related to death included septic shock/multiple organ failure (18 cases, 50%), respiratory failure (8 cases, 22%), brain death (4 cases, 11%), non-septic shock (3 cases, 8%), and hepatocellular failure (3 cases, 8%) (Table 2). The post-mortem microbiological study demonstrated the presence of yeast in 16/36 autopsies (44.4%). In eight of them, only positive cultures for yeast were obtained; in four cases, the presence of yeast was only demonstrated in histology; and in another four cases, yeasts were present in both the cultures and in histology. The organs affected according to the culture, histology, or both simultaneously are detailed in Table 3. The species isolated in the post-mortem cultures were *Candida albicans* in nine cases, *Candida glabrata* in two cases, and *Candida tropicalis* in one case. There were three patients in whom the result of the post-mortem study did not coincide with the pre-mortem study. In two cases, this was because the diagnosis was made post-mortem and, in the remaining case, the species isolated in the post-mortem study was different from that identified in the patient in life. In the remaining nine cases, the species identified in the post-mortem study matched the one obtained in the follow-up of the living patient.

Attributable mortality:

Attributable mortality according to the post-mortem study: Of the 16 patients in whom yeast was identified in the post-mortem study, there were 6 who died from respiratory failure, 5 from septic shock or MODS, 2 from non-septic shock, 2 from brain death, and 1 from hepatocellular insufficiency. According to the criteria based on the post-mortem study, death was attributed to yeast in 11/36 cases (30.6%).

Attributable death based on the clinical study: Of the 31 cases in which no post-mortem study was performed, 7 cases (22.6%) were considered to meet the criteria for death attributable to yeast.

Crude death rate: The attributable post-mortem mortality along with the attributable clinical mortality represents 26.9% (18/67) of deaths in the studied group.

Attributable mortality according to statistical analysis: If the ICU mortality of the studied population, from which *Candida* spp. was isolated during their stay in the ICU, was 35% (51/145) and that of the global population treated in the ICU during the study was 9.6% (324/3389), the statistically attributable mortality is 25%. However, if we compare only the mortality of the study group (145 patients), grouped according to the mortality of patients with IC (51%; 61/120) vs. the CC group (24%; 6/25), the resulting statistically attributable mortality is 27%.

Predictive factors for yeast-attributable mortality are shown in
Table 4, including age, previous illness, APACHE III score, abdominal surgery on admission, positive urine culture at follow-up, and identification of *Candida glabrata* and *Candida tropicalis* at screening. Through multivariate analysis, three variables with statistical significance were selected: abdominal surgery on admission, *Candida glabrata* at screening, and appropriate antifungal treatment as a protective factor (Table 5).

## 4. Discussion

In the 1980s, in non-neutropenic critically ill patients in the ICU [1], the mortality rate from DC was >80% and that from IC was >50% [1,24]. The diagnostic algorithm has allowed for early antifungal treatment and a significant reduction in mortality in this population affected by candida infection.

A few years later, it was shown that incorporating MC into the IC group within the diagnostic algorithm reduced mortality and incidence of DC. These results could not have been achieved using the criteria agreed upon in the EORTC guidelines, which are the same as those used in the FUNDICU Consensus published in 2024 [2,25,26]. This is because, under those criteria, CM and CC are considered equivalent and are not adequately adjusted for non-neutropenic critically ill patients. The ICU population with multiorgan dysfunction has no relation to neutropenic or transplanted patients. The presented algorithm and other alternatives can be chosen, such as the Candida Score presented by the EPCAN Group [27] or the *Candida* Colonization Index presented by the Pittet Group [28,29], with a single objective: the reduction in attributable mortality to *Candida* through early antifungal treatment.

Applying this algorithm resulted in attributable mortality of 30.6%, according to the post-mortem study. In similar studies where the number of post-mortem studies exceeds 50% of the deceased patients, the determined attributable mortality ranges between 47% and 75% [30,31,32,33,34]. In other reports in which the number of post-mortem studies performed was not recorded, the attributable mortality determined post-mortem ranged between 38% and 52% [20,35,36,37,38,39,40].

After 45 years, it can be stated that the present algorithm is of great utility in significantly reducing the attributable mortality to *Candida* in non-neutropenic critically ill patients in the ICU [41]. This is an important improvement given that this is a serious infection acquired in the ICU, mainly due to endogenous colonization. This algorithm is designed to identify the population at risk of DC in the ICU and agrees with the three points proposed by Ergün M et al. [42] regarding the algorithm to be developed: host factors, clinical factors, and mycological evidence.

Once the group of patients with candidiasis susceptible to antifungal treatment has been identified, following this algorithm, it is of interest to use the attributable mortality indicator as a metric for monitoring therapeutic results. This indicator provides the best data that can help to detect areas of improvement in diagnostic and therapeutic decision-making with the aim of improving patient outcomes. It takes into account the effect of infection on mortality and the prevalence of the infection within the target population [4].

Univariate statistical analysis was performed, and it was found that previous illness, age, APACHE III score (Acute Physiology Age and Chronic Health Assessment is a prognostic scoring system in ICU) [43], and abdominal surgery are risk factors in relation to attributable mortality. The existence of more than one focus and a positive urine culture for yeast in the follow-up study are also risk factors for attributable mortality.

Multivariate analysis shows that abdominal surgery and isolation of *non-albicans Candida* (*Candida glabrata*) are risk factors for attributable mortality, while appropriate antifungal treatment is a protective factor.

In this study, attributable mortality to *Candida* was 25–27%, while the crude death rate was 26.9%. These results indicate that the selection of patients at risk of IC is correct. At the same time, the attributable mortality indicates that 25–27% of these patients have not received adequate antifungal treatment. This analysis encouraged a review of the therapeutic algorithm, suggesting that therapeutic individualization should be carried out. This involves adjusting the initial antifungal treatment based on the identification of a yeast with a high probability of antifungal resistance (e.g., *C. glabrata* and *C. krusei*). For example, DC demonstrated in the first sample may lead to poorly tolerated sepsis or severe liver failure. Therapeutic changes have resulted in a statistically attributable mortality to IC of 0% (35% (21/60) IC mortality vs. 38% (16/42) CC mortality) and a crude death rate of 4.8% (1/21) [44].

The change in the therapeutic protocol, resulting from the analysis of statistically attributable mortality data, did not modify the incidence of IC in critically ill patients treated in the ICU (4.3% in this study, 3.5% described in EPIC study [45], and 3.2% in the group of patients undergoing the new therapeutic protocol) [44]. However, it did significantly modify the incidence of disseminated candidiasis (26% (31/120) vs. 7% (4/60), *p* = 0.002) and attributable mortality (30.6% (11/36) vs. 0% (0/8)) to candidemia (14% (17/120) vs. 5% (3/60)) and endophthalmitis (2.5% (3/120) vs. 0% (0/60)), according to the post-mortem study. According to the review by Eggimann Ph. [46], the incidence of candidemia in these patients ranged between 10 and 20%, and that of endophthalmitis ranged between 3.7 and 25%.

The diagnostic algorithm, while maintaining adequately identified host factors and clinical factors, can be adapted to possible improvements by incorporating new technologies that allow for faster detection of mycological infection, as long as they do not delay antifungal therapeutic-related decision-making or lead to unnecessary antifungal treatments. The antifungal therapeutic protocol that is activated with this diagnostic algorithm will have to be adjusted for the detection of new species resistant to antifungals, for the incorporation of new antifungals with fewer side effects, and for the costs of these treatments. Follow-up with statistically attributable mortality is significant to evaluate the changes that can be made and for the continuous improvement in the care of these ICU patients.

## 5. Conclusions

This study validated the concordance of results regarding attributable mortality due to *Candida* spp. in critically ill ICU patients, based on three distinct methods: statistical analysis, post-mortem studies, and clinical studies.

The use of this metric may help to improve IC care outcomes in non-neutropenic critically ill patients in the ICU.

## Figures and Tables

**Figure 1 jof-11-00871-f001:**
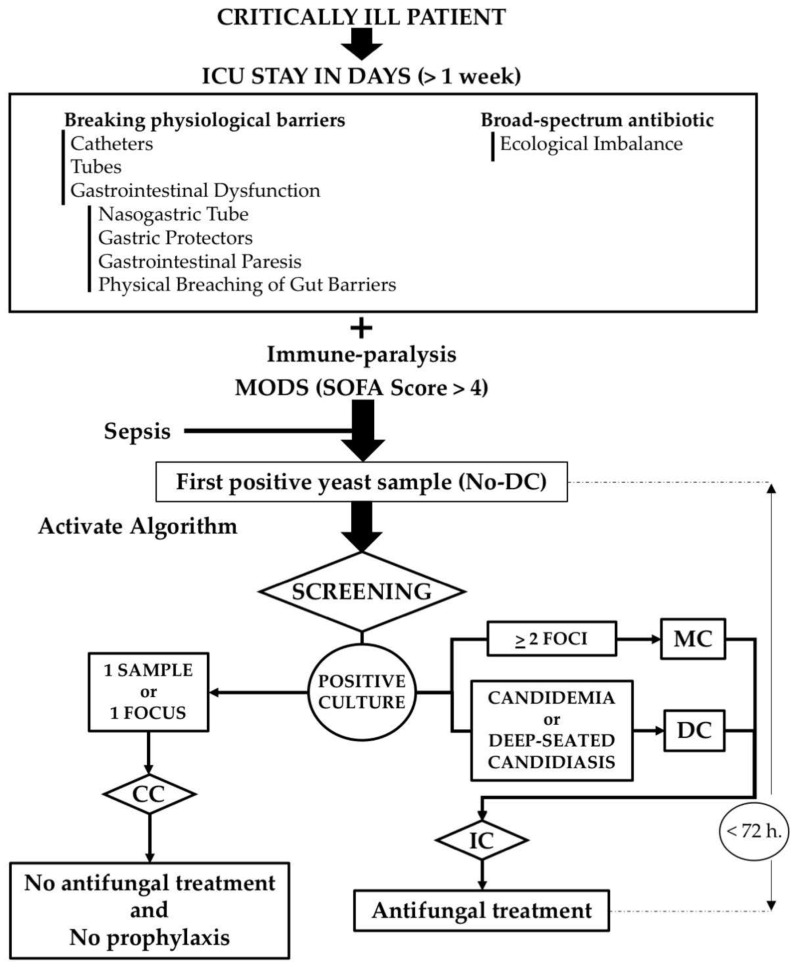
Diagnostic-therapeutic algorithm applied CC = *Candida* spp. Colonization; MC = Multifocal Candidiasis; DC = Disseminated Candidiasis; IC = Invasive Candidiasis; ICU = Intensive Care Medicine; MODS = Multiple Organ Dysfunction Syndrome. Definitions: MC: Simultaneous isolation of *Candida* spp. in two or more foci: respiratory (bronchial secretions in intubated patients), digestive (gastric aspirated plus throat smear), urinary (in catheterized patient), or deep wounds (by puncture or drainage). DC: Microbiological evidence of yeasts in fluids from normally sterile sites (cerebrospinal fluid, pleural or pericardial fluids, peritoneal or bile fluids) or histological samples from deep organs, endophthalmitis, or candidemia with negative line-tip cultures.

**Figure 2 jof-11-00871-f002:**
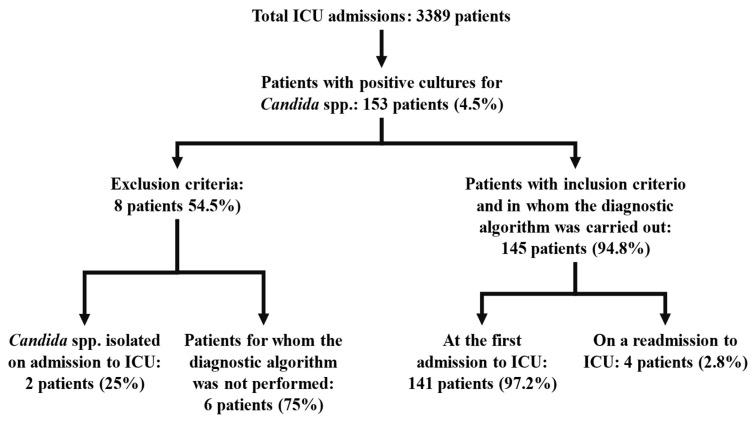
Patients with *Candida* spp. isolation during ICU stay.

**Table 1 jof-11-00871-t001:** Presence of risk factors for candidiasis in the study group.

Risk Factors for Candidiasis	Number of Cases (%)
Antibiotics	144 (99)
Central venous catheter	143 (99)
Urinary catheter	140 (97)
Antacid therapy	141 (97)
Naso-orogastric catheter	131 (90)
Arterial catheter	125 (86)
Oro-nasotracheal tube and/or tracheostomy	124 (85)
Surgical procedures	85 (59)
Vasoactive drugs	72 (50)
Total parenteral nutrition	68 (47)
Drainages	65 (45)
Blood derivatives	62 (43)
Corticoids	56 (39)
Hemodyalisis	8 (5)
Splenectomy	5 (3)

**Table 2 jof-11-00871-t002:** Causes of death in the study group.

Causes of Death	Number of Cases (%)
Septic shock–Multiple Organ Dysfunction Syndrome	18 (50)
Respiratory failure (hypoxemia)	8 (22)
Brain death	4 (11)
Non-septic shock	3 (8)
Hepatic failure	3 (8)

**Table 3 jof-11-00871-t003:** Post-mortem evidence of yeasts of the study group.

Location	Positive Cultures Only	Positive Histology Only	Positive Cultures and Histology	All Positive Cultures (n = 24) (%)	All Positive Histology (n = 36) (%)
Lung	5	3	3	8 (33)	6 (17)
Trachea	4			4 (17)	
Bowel	2	1	1	3 (12)	2 (6)
Heart	1	2		1 (4)	2 (6)
Kidney	1	2		1 (4)	2 (6)
Urinary tract	3			3 (12)	
Liver	2			2 (8)	
Spleen	2			2 (8)	
CNS		1			1 (3)
Positive samples	20	9	4	24	13

CNS: Central nervous system.

**Table 4 jof-11-00871-t004:** Characteristics of patients by attributable mortality.

Variables	NAM	AM	P
Age	51 (22)	69 (14)	0.001 *
Previous illness	2 (0–4)	3 (1–4)	0.009 †
Apache III	72 (24–136)	90 (58–183)	0.040 †
Abdominal surgery on admission	24/128 (19)	9/17 (53)	0.001 ‡
More than one *focus* (risk classification)	100/128 (78)	17/17 (100)	0.043 §
*Candida glabrata* at screening	20/128 (16)	6/15 (40)	0.020 ‡
*Candida tropicalis* at screening	6/128 (5)	3/15 (20)	0.036 §
Urine cultures positive at follow-up	20/128 (16)	13/75 (75)	<0.001 §

* Student’s *t* test. Mean (SD). † U Man-Whitney test. Median (range of values). ‡ Chi-square test. Number of positive cases/total (%). § Fisher Exact test. Number of positive cases/total (%). NAM: Non-attributable mortality. AM: Attributable mortality.

**Table 5 jof-11-00871-t005:** Adjusted Odds Ratio of attributable mortality for variables of interest.

Variables	Attributable Mortality		
	Odds Ratio	95% Confidence Interval	*p*
Abdominal surgery on admission	9.17	1.77–47.39	0.008
Appropriate antifungal treatment	<0.01	<0.01–0.10	<0.001
*Candida glabrata* at screening	7.38	1.24–43.98	0.028
Previous illness	1.82	0.98–3.37	0.055

## Data Availability

The original contributions presented in this study are included in the article. Further inquiries can be directed to the corresponding author.

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
