# Peer review of "Attributable Mortality to Candidiasis in Non-Neutropenic Critically Ill Patients in the ICU and a Post-Mortem Study"

_jof, 2025, doi:10.3390/jof11120871_

Round 1
Reviewer 1 Report
The manuscript called “Attributable Mortality from Candidiasis in Non-Neutropenic Critically Ill Patients in the ICU and a Post-Mortem Study” provides a thorough, systematic, and methodologically sound investigation into how candidiasis affects mortality rates in a high-risk patient group. It employs a prospective study design, a detailed diagnostic approach, and post-mortem analyses. This research is very important because it helps fill a significant knowledge gap about how much candidiasis contributes to mortality. By using the concept of attributable mortality, it quantifies the public health impact of candidiasis and offers valuable information for making informed decisions about healthcare resource allocation. The study also presents a custom diagnostic and treatment plan for early detection and management of candidiasis. This could be a practical tool for doctors and may help reduce death rates related to candidiasis by offering a more comprehensive approach to treatment. Additionally, using post-mortem studies to verify clinical findings enhances the accuracy of conclusions about mortality. This provides a solid foundation for future research and the development of clinical practice guidelines.
However, the study does have some limitations:
1. The number of patients with candidiasis in the study is quite small (only 145 cases), which limits the statistical power of the study and how broadly its findings can be applied. Collaborating with multiple centers to increase the sample size would make the results more generally applicable and include a more diverse group of patients.
2. There is a lot of variation in how antifungal treatments are administered, including which drugs are used and their duration. This can affect the results and make it difficult to determine the best treatment strategies. Standardizing these protocols based on evidence and expert consensus would make it easier to compare outcomes.
3. Since the study was conducted at only one center, its findings might not apply to other ICUs. Conducting studies across multiple centers would improve this, and follow-up studies to track patient outcomes over time could provide more insight into how effective and enduring the intervention is, which would help guide future improvements.
Overall, the manuscript makes a valuable contribution to the field. However, if the authors address the issues we've highlighted and incorporate the suggested improvements, the study could become even better, more widely applicable, and have a greater impact on clinical practice.
Author Response
- The number of patients with candidiasis in the study is quite small (only 145 cases), which limits the statistical power of the study and how broadly its findings can be applied. Collaborating with multiple centers to increase the sample size would make the results more generally applicable and include a more diverse group of patients
- It is true that the number is not high because this is a postmortem study due to the difficulty of performing autopsies. A multicentre study could be considered
- There is a lot of variation in how antifungal treatments are administered, including which drugs are used and their duration. This can affect the results and make it difficult to determine the best treatment strategies. Standardizing these protocols based on evidence and expert consensus would make it easier to compare outcomes.
- The standardisation of antifungal treatment should enable comparison between them
Reviewer 2 Report
Dear Authors
thank you very much for this work.
It is an interesting study that clearly contributes to the subject matter and proposes a certain correction when addressing deaths in non-neutropenic patients due to Candida species.
Several suggestions are included in the PDF file. Remember that “spp” should not be italicized, only the genus and species name, where applicable.
Kind regards
In the PDF file are highlighted all the changes proposed.

Author Response
We have reviewed the points you mentioned, removing any incorrect notes and correcting all those you indicated
Suggested changes in yellow

Reviewer 3 Report
The study addresses the quantification of the real mortality burden of Candida infections in ICU settings, which is an important clinical problem. The reviewer acknowledges the significant work performed by the authors, but certain aspects need to be improved substantially. While the text is dense, the introduction fails to provide a comprehensive overview and lacks the definition and explanation of several terms that are fundamental to the article. The results sections are presented almost as bullet points of facts instead of a text that has a logical connection between each result. The study design and its possible biases are not properly described in the methods section. The use of colloquial terms such as "one wonders" and "one can agree and disagree" is not appropriate for scientific writing. Therefore, I'd recommend a thorough review and rewrite of the text.
- Abstract lacks a proper introduction of the subject and jumps straight into stating the study's purpose.
- The first chapter of the introduction should introduce Candida more thoroughly. Additionally, the use of sentences such as "Monto Ho wrote" and quotation marks for sentences of an article are not commonly used in scientific writing, and I'd suggest the text be revised (Line 28).
- Line 41: Similarly, the sentence "One wonders" is not the most appropriate way to pose scientific questions and hypotheses.
- Lines 43 - 49: The definition of attributable mortality and crude mortality, their difference, and why one is preferable could be more thoroughly explained
- IC is used both in the abstract and introduction (Line 51) without a definition of what it means. I also suggest that it is further explored in the introduction what the difference is between IC and MC, and why there is debate around it.
- Lines 71 - 72: A brief explanation of the diagnostic algorithm should be given in the text.
- Lines 80 - 85: These definitions should be given in the text, not on a figure legend.
- If the result section is broken into subsections, I'd suggest giving them titles that summarize that section's conclusion, instead of having "demographics:", "antifungal treatment:"
- Line 198: "average age of 53 (22) years". What does the 22 mean? The range between ages? This should be better described in the results.
- Line 199-200: I'd suggest explaining what the APACHE III is and what the number represents.
- Line 210: In the Fungal Infection results section, I'd suggest describing it in paragraph form, and not in bullet points. Same thing in the Attributable Mortality section.
- Line 214-220: The whole description of the Blood Cultures is confusing. The use of "first culture", "screening", and follow-up cultures" confused me. Is this explained in the methods?
- In the intro, the author mentions that this work can help overcome the debate of whether MC and IC can be equated, but in the results, the authors already use MC and IC as if they are the same thing
- The use of colloquial speech in the discussion is not appropriate for scientific writing
- If the study predates the use of echinocandins, maybe that could be discussed
- I'd suggest also discussing the limitations of a single-center study
Author Response
Abstract lacks a proper introduction of the subject and jumps straight into stating the study purpose.
The first chapter of the introduction should introduce Candida more thoroughly. Additionally, the use of sentences such as Monto Ho wrote and quotation marks for sentences of an article are not commonly used in scientific writing, and
suggest the text be revised
We have reviewed the points you mentioned, removing any incorrect notes and correcting all those you indicated (changes in yellow)
